



**Responses of surface ozone to future agricultural ammonia emissions**
**and subsequent nitrogen deposition through terrestrial ecosystem**
**changes**
Xueying Liu[1*], Amos P. K. Tai[1,2,3], Ka Ming Fung[1#]
[1]Earth System Science Programme and Graduate Division of Earth and Atmospheric Sciences, Faculty of Science, The Chinese
University of Hong Kong, Sha Tin, Hong Kong SAR, China
[2]Institute of Environment, Energy and Sustainability, and State Key Laboratory of Agrobiotechnology, The Chinese University
of Hong Kong, Sha Tin, Hong Kong SAR, China
[3]Center for Soybean Research of the State Key Laboratory of Agrobiotechnology, The Chinese University of Hong Kong,
Hong Kong SAR, China
[*]Now at: Department of Earth and Atmospheric Sciences, University of Houston, Houston, TX, USA
[#]Now at: Department of Civil and Environmental Engineering, Massachusetts Institute of Technology, Cambridge, MA,
USA
*Correspondence to*: Amos P. K. Tai (amostai@cuhk.edu.hk)
**Abstract.** With the rising food demands from the future world population, more intense agricultural activities are expected to
cause substantial perturbations to the global nitrogen cycle, aggravating surface air pollution and imposing stress on terrestrial
ecosystems. Much less studied, however, is how the terrestrial ecosystem changes induced by agricultural nitrogen deposition
may modify biosphere-atmosphere exchange and further exert secondary feedback effects on global air quality. Here we
examined the responses of surface ozone air quality to terrestrial ecosystem changes caused by 2000-to-2050 changes in
agricultural ammonia emission and the subsequent nitrogen deposition by asynchronously coupling between the land and
atmosphere components within the Community Earth System Model framework. We found that global gross primary
production is enhanced by 2.1 Pg C yr$^{-1}$ following a 20% (20 Tg N yr$^{-1}$) increase in global nitrogen deposition by the end of
year 2050 in response to rising agricultural ammonia emission. Leaf area index was simulated to be higher by up to 0.3–0.4
m$^2$ m$^{-2}$ over most tropical grasslands and croplands, and 0.1–0.2 m$^2$ m$^{-2}$ across boreal and temperate forests at midlatitudes.
Around 0.1-0.4 m increases in canopy height were found in boreal and temperate forests, and ~0.1 m increases in tropical
grasslands and croplands. We found that these vegetation changes could lead to surface ozone changes by ~0.5 ppbv when
prescribed meteorology was used (i.e., large-scale meteorological responses to terrestrial changes were not allowed), while
surface ozone could typically be modified by 2–3 ppbv when meteorology was dynamically simulated in response to
vegetation changes. Rising soil NO$_x$ emission from 7.9 to 8.7 Tg N yr$^{-1}$ could enhance surface ozone by 2–3 ppbv with both
prescribed and dynamic meteorology. We thus conclude that following enhanced nitrogen deposition, the modification of the
meteorological environment induced by vegetation changes and soil biogeochemical changes are the more important pathways
that can modulate future ozone pollution, representing a novel linkage between agricultural activities and ozone air quality.
**1 Introduction**
Increased food production for the ever-growing world population has been enabled by the widespread agricultural expansion
and intensification with heavy fertilizer applications, which have correspondingly led to an enhancement in ammonia (NH$_3$)
emission from the land by a factor of two to five since preindustrial times (Behera et al., 2013; Gu et al., 2015; Zhu et al.,
2015). For instance, Asia (excluding Siberia), home to more than 60% of the world population (FAOSTAT, 2016), has
experienced rapid expansion of agricultural activities (Liu & Tian, 2010; Tian et al., 2014), accounting for ~50% of the global
total consumption of synthetic fertilizer and 30–40% of global manure production (FAOSTAT, 2016). Agriculture-related
activities are known to be the most significant sources of atmospheric NH$_3$, of which the vast majority (~60%) originates from



the excessive use of nitrogenous fertilizer and concentrated operations of livestock feeding on a global scale (Huang et al.,
2012; Paulot et al., 2014; Zhang et al., 2018); for Asia the percentage is even higher (80–90%) (Streets et al., 2003; Reis et al.,
2009; Gu et al., 2012; Kang et al., 2016; Zhang et al., 2017; Zhang et al., 2018). Crops typically take up only about 40–60%
of the fertilizer nitrogen applied to croplands (Tilman et al., 2002; Zhang et al., 2015; Liu et al., 2016; Muller et al., 2017), and
only 25–35% of the nitrogen fed to dairy cows is converted into milk (Bittman et al., 2009), while most of the remainder is
chemically transformed into a variety of simple and complex forms and leaked to the environment. The release of gaseous
$NH_3$ into the atmosphere is one of the major nitrogen leakages from agricultural soils. Under a business-as-usual scenario
where future nitrogen use efficiency (NUE; i.e., the fraction of nitrogen input finally harvested as output) in agricultural
systems is not expected to be substantially improved, increasing food production will undoubtedly continue to intensify
agricultural $NH_3$ emission into the overlying air (Erisman et al., 2008; Lamarque et al., 2011; Zhang et al., 2017).
Reactive nitrogen, from emissions of nitrogen oxides ($NO_x$; $NO+NO_2$) and $NH_3$, is deposited over land and ocean through a
variety of processes collectively known as wet and dry deposition. As combustion-driven $NO_x$ emission is projected to slow
down due to regulatory efforts (van Vuuren et al., 2011) while agricultural $NH_3$ emission will continue to increase (Lamarque
et al., 2011), future nitrogen deposition is expected to increase overall in the global budget (Galloway et al., 2004; Paulot et
al., 2013; Lamarque et al., 2013; Kanakidou et al., 2016) and shift from a nitrate-dominated to ammonium-dominated condition
(Ellis et al., 2013; Paulot et al., 2013; Li et al., 2016). Atmospheric nitrogen deposition onto the land surface is an important
source of soil mineral nitrogen and thus enhances plants growth; this is known as the "nitrogen fertilization effect" (Reay et
al., 2008; Templer et al., 2012). The fertilization effect depends on the soil "nitrogen limitation" defined as the nitrogen
constraint on the productivity of many terrestrial ecosystems (Vitousek et al., 2002; Gruber and Galloway, 2008; LeBauer et
al., 2008; Heimann et al., 2008; Reay et al., 2008; Zaehle et al., 2010). Nitrogen limitation is often found in natural soils where
severe nitrogen competition among plants and microbes exists, and the unmet plant nitrogen demand can be translated to a
reduction in the potential gross primary production (GPP) of the terrestrial ecosystems, representing a direct downregulation
of photosynthetic carbon gain.

Nitrogen deposition affects the terrestrial carbon and nitrogen cycle, but much less is known about how nitrogen deposition
affects atmospheric chemistry via terrestrial changes and feedbacks. As nitrogen limitation is relaxed, enhanced carbon
assimilation can be translated to changes in the carbon mass allocated to different plant parts, ultimately manifested as an
enhancement in vegetation structural variables such as leaf area index (LAI) and canopy height. Meanwhile, nitrogen
deposition can also alter soil inorganic nitrogen composition and a variety of abiotic and biotic processes including uptake by
plants, nitrification, denitrification, immobilization by microbes, and fixation in clay minerals. Soil $NO_x$ is produced as a by-
product of nitrification and denitrification, two microbial processes that first convert $NH_3$ aerobically to nitrate ($NO_3^-$) and
then $NO_3^-$ to nitrous oxide ($N_2O$) or nitrogen gas ($N_2$) under anoxic conditions. As LAI, canopy height and soil $NO_x$ are known
to affect surface air quality, nitrogen deposition can potentially affect atmospheric chemistry through affecting vegetation
structure and ecophysiology, as well as soil biogeochemistry.

Nitrogen-mediated changes in vegetation and soil can affect surface ozone air quality via various pathways (Fig. 1). Among
them, "biogeochemical" effects are processes mediated via direct exchange (i.e., emissions or deposition) of relevant chemical
species between the terrestrial biosphere (vegetation and soil microbes) and the atmosphere, while "biogeophysical" or
"meteorological" effects are mediated through a modification of the overlying meteorological environment (i.e., temperature,
humidity, turbulence structure, etc.), as defined in Sadiq et al. (2017), Zhou et al. (2018) and Wang et al. (2020). One possible
biogeochemical pathway is that LAI enhancement could elevate surface ozone by increasing biogenic volatile organic
compound (VOC) emissions in high-$NO_x$ environments, but could also reduce ozone by increasing dry-depositional uptake





via leaf stomata (Zhao et al., 2017). Another possible biogeochemical effect is via the increase in canopy height, which further
enhances surface roughness length, turbulent mixing and thus higher aerodynamic conductance for land-atmosphere exchange
including ozone dry deposition (Bonan, 2016; Oleson et al. 2013). Another possible biogeochemical effect is that increased
inorganic nitrogen availability facilities soil $NO_x$ emission through nitrification and denitrification processes, which further
causes rapid NO and $NO_2$ cycling for ozone formation. Biogeophysical effects or meteorological effects are through
vegetation-induced changes in the surface energy balance (e.g., absorbed solar radiation, sensible and latent heat fluxes) and
subsequent changes in surface temperature, precipitation, humidity, circulation patterns, moisture convergence (Wang et al.,
2020). Higher temperature enhances ozone mainly through increased biogenic emissions and higher abundance of $NO_x$, while
lower humidity reduces the chemical loss rate of ozone (Jacob and Winner, 2009; Fiore et al., 2012). Surface ozone changes
via each individual process are heterogeneous over the globe, and the overall ozone response through various biogeochemical
and biogeophysical pathways is highly complex (Zhao et al., 2017).

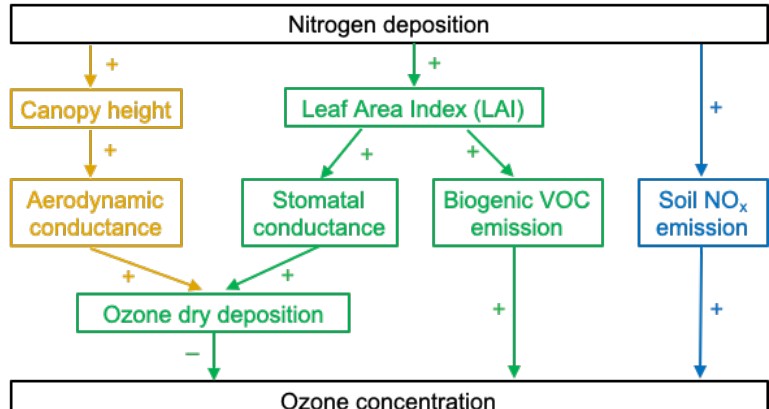

**Figure 1.** "Biogeochemical" pathways of nitrogen deposition affecting surface ozone concentration. Pathways via canopy
height (yellow-colored), leaf area index (LAI; green-colored) and soil $NO_x$ (blue-colored) are shown. The sign associated with
each arrow indicates the correlation between the two variables; the sign of the overall effect (positive or negative) of a given
pathway is the product of all the signs along the pathway. "Biogeochemical" pathways affect gas exchange (i.e. biogenic VOC
emission and ozone deposition) though plant stomata or microbe-mediated soil processes.
Here we present a study that investigates how agriculture-induced increases in $NH_3$ emission and subsequent nitrogen
deposition could affect surface ozone air quality via terrestrial ecosystem changes in terms of LAI, canopy height and soil $NO_x$
emission. We used an asynchronously coupled modeling framework based on the atmosphere (CAM-Chem) and land (CLM)
components of the Community Earth System Model (CESM) to quantify the corresponding responses of surface ozone air
quality to terrestrial changes. We first examined the responses of vegetation and soil variables to the present-day vs. future
scenarios of nitrogen deposition and then use those terrestrial changes to drive factorial simulations for surface ozone. To
evaluate the relative importance of LAI, canopy height and soil $NO_x$ emission, we evaluated ozone responses to the three
individual effects and the overall combined effects using prescribed meteorology (i.e., large-scale meteorological responses to
terrestrial changes are not allowed). Furthermore, we evaluated the effects of changing meteorology to surface ozone by
conducting simulations using dynamic meteorology (i.e., where overlying boundary-layer meteorology and large-scale
circulation also responds to terrestrial changes). Model configuration with dynamic meteorology represents the overall effects
from regional terrestrial changes and associated meteorological changes (an integration over both biogeochemical and
biogeophysical effects to surface ozone), whereas the setting with prescribed meteorology provides limited above-surface layer



meteorological changes directly caused by terrestrial changes and represents the biogeochemical effects only. Our study
emphasizes the complexity of biosphere-atmosphere interactions and their indirect modulating effects on air quality and
atmospheric chemistry, which are important for evaluating the impacts from future food production trends on air quality and
health beyond the direct effects of agricultural emissions alone.
**2 Model and Method**
**2.1 Model description**
We used the Community Earth System Model (CESM), which includes atmospheric, land, ocean and sea ice model
components. We employed CESM version 1.2.2 with fully interactive atmosphere and land components, but with prescribed
ocean and sea ice consistent. For the atmosphere component, we used the Community Atmosphere Model version 4 (CAM4)
(Neale et al., 2010) fully coupled with an atmospheric chemistry scheme (i.e., CAM-Chem) that contains full tropospheric $O_3$-
$NO_x$-CO-VOC-aerosol chemistry based on the MOZART-4 mechanism (Emmons et al., 2010; Lamarque et al., 2012).
Emissions are from the combined emission inventories of the Emissions Database for Global Atmospheric Research
(EDGAR), Regional Emission inventory in ASia (REAS) and Global Fire Emissions Database (GFED2) and others. CAM-
Chem provides the flexibility of performing climate simulations online (i.e., "dynamic meteorology") and simulations with
specified meteorological fields (i.e., "prescribed meteorology"). For simulations with dynamic meteorology, it was driven by
the Climatic Research Unit – National Centers for Environmental Prediction (CRU-NCEP) climate forcing dataset. For
simulations with prescribed meteorology, year-2000 and 2001 horizontal wind components, air temperature, surface
temperature, surface pressure, sensible and latent heat flux and wind stress of the Goddard Earth Observing System Model
version 5 (GEOS-5) forcing data at six hour interval were used (see Table 1). This version of CAM-Chem simulates the
concentrations of 56 atmospheric chemical species at a horizontal latitude-by-longitude resolution of 1.9°×2.5° and a vertical
resolution of 26 layers for dynamic meteorology and 52 layers for prescribed meteorology.

For the land component, we used the Community Land Model version 4.5 (CLM4.5) (Oleson et al., 2013) with Satellite
Phenology (CLM45SP) mode where vegetation structures are prescribed (e.g., using satellite-derived LAI data), or with active
carbon–nitrogen biogeochemistry (CLM45BGC) that contains prognostic treatment of terrestrial carbon and nitrogen cycles
(Lawrence et al., 2011), depending on the cases of concern. In CLM4.5, the Model of Emissions of Gases and Aerosols from
Nature (MEGAN) version 2.1 was used to compute biogenic emissions online as functions of LAI, vegetation temperature,
solar radiation, soil moisture and other environmental conditions (Guenther et al., 2012). For dry deposition of gases and
aerosols we used the resistance-in-series scheme in CLM4.5 as described in Lamarque et al. (2012) with updated, optimized
coupling of stomatal resistance to LAI (Val Martin et al., 2014). Soil $NO_x$ emission was implemented by Fung et al. (2021) as
a function of $N_2O$ emission, soil air-filled pore space and volumetric soil water content during nitrification and denitrification
(See Supplementary for details). We also applied a temperature factor to correct the soil $NO_x$ overestimation at high latitudes
as previous studies (Zhao et al., 2017). Evapotranspiration rate was calculated based on the Monin-Obukhov similarity theory
for turbulent exchange and the diffusive flux-resistance model with dependence on vegetation, ground and surface temperature,
specific humidity, and an ensemble of resistances that are functions of meteorological and land surface conditions (Oleson et
al., 2013; Lawrence et al., 2011; Bonan et al., 2011).



## 2.2 Asynchronously coupled atmosphere chemistry-biosphere modeling framework

An asynchronously coupled system with CAM-Chem and CLM was adopted to investigate the vegetation structural changes induced by nitrogen deposition and their potential to modulate surface ozone under both dynamic and prescribed meteorology. Asynchronous instead of synchronous coupling was used because currently CESM does not have the capacity to allow "online" bidirectional exchange of reactive nitrogen fluxes between the atmosphere and land components; it also conveniently facilitates sensitivity experiments to be conducted to isolate individual drivers of changes and processes. First, present-day and future scenarios of nitrogen deposition are obtained by CAM-Chem simulations with the corresponding $NH_3$ emission of year 2000 and 2050. Year-2000 $NH_3$ emission was from the prescribed emission inventory inherent in CAM-Chem (see Sect. 2.1), which includes anthropogenic, ocean, soil and biomass burning sources. We split the year-2000 anthropogenic $NH_3$ emission into agricultural and non-agricultural parts by using the corresponding ratios based on the Magnitude And Seasonality of Agricultural Emissions model for $NH_3$ (MASAGE_NH3) (Paulot et al., 2014). We kept natural and non-agricultural emissions the same in both the year-2000 and year-2050 scenarios, and only scaled the year-2000 agricultural $NH_3$ by a growth factor $g$ (Fig. 1c)

$$g = \frac{\text{crop production in 2050}}{\text{crop production in 2000}} \qquad \text{Eq.1}$$

based on crop production estimates from Alexandratos and Bruinsma (2012) accounting for technology-driven yield improvements and cropland area changes, as in Tai et al. (2014; 2017), and weighted by crop-specific production growths. Such a linear scaling assumes nitrogen-use efficiency (NUE) of fertilization applications to remain the same in the future. In practice NUE is expected to rise with technological advancements, the extent of which is however highly uncertain and region-specific; we therefore regarded our linear scaling as a representation of the "worst-case" scenario where fertilizer nitrogen use remains as inefficient as it is today. For each scenario of the sensitivity experiments, simulations were conducted for 20 simulation years, and the first five years of outputs were treated as spin-up and thus discarded in the analysis. We calculated the annual averages of the last 15 years to obtain the corresponding nitrogen deposition fluxes for the year-2000 and year-2050 scenarios. Throughout the CAM-Chem component was still coupled online with CLM45SP mode with prescribed vegetation structures, which computes biogeophysical fluxes that CAM-Chem requires for atmospheric dynamics and chemistry; i.e., asynchronous coupling mentioned above was only referring to nitrogen fluxes.

The CLM45BGC mode was used to investigate vegetation and soil changes in response to perturbations in the nitrogen input to the land. We first obtained steady-state vegetation and soil variables including LAI, canopy height and soil $NO_x$ emission following present-day nitrogen deposition (obtained above CAM-Chem) for 200 years in CLM. The first 150 years of outputs were treated as spin-up and thus discarded in the analysis, while the last 50-year average was used to represent the vegetation and soil conditions in a steady state. We then perturbed the present-day steady state with future nitrogen deposition fluxes following the year-2050 agricultural emission scenario, allowing the vegetation and soil variables to come into a "new" steady state, which took 10–20 simulations years. After that, the simulation was conducted for another 50 years, which were then averaged to determine the differences in LAI, canopy height and soil $NO_x$ emission from the 50-year present-day averages.

Last, we investigated the individual and combined impacts of the above changes in the three terrestrial pathways (i.e., via LAI, canopy height and soil $NO_x$ emission) on surface ozone air quality, with both prescribed meteorology (i.e., large-scale meteorological responses to terrestrial changes are not allowed) and dynamic meteorology (i.e., overlying boundary-layer meteorology and large-scale circulation also responds to terrestrial changes). Terrestrial changes with prescribed meteorology included only biogeochemical pathways, while terrestrial changes with dynamic meteorology included the combined effects of biogeochemical and biogeophysical processes as well as larger meteorological and circulation pattern changes. Therefore, we were able to examine the effects from land-atmosphere feedbacks with dynamic meteorology, while prescribed





meteorology provided limited atmospheric changes directly caused by terrestrial changes without much land-atmosphere
feedbacks. To evaluate the relative importance of individual pathways to the overall effects, we conducted four sets of fully
coupled land-atmosphere simulations: (1) a control case without any nitrogen-mediated changes in LAI, canopy height and
soil $NO_x$ emission ([CTR]); (2) a simulation with LAI change only ([LAI]); (3) a simulation with canopy height change only
([HTOP]); (4) a simulation with soil $NO_x$ emission change only ([NOX]); (5) a simulation with all changes in LAI, canopy
height and soil $NO_x$ emission ([ALL]). Simulation [LAI], [HTOP] and [NOX] in relation to [CTR] allowed us to quantify the
relative contribution from LAI, canopy height and soil $NO_x$ emission respectively, while simulation [ALL] reflected the overall
ozone changes due to three combined effects. The experiments were summarized in Table 1. We conducted the same set of
simulations with both dynamic and prescribed meteorology to examine how meteorological responses to these terrestrial
changes would modify the importance of these pathways (Table 2). We focused on average changes in the last 15-year northern
summer (June, July and August: JJA) for most of the variables in the rest of this paper, since summer was both the high-ozone
season and the growing season of the majority of global vegetation, when ozone-vegetation coupling appeared to be the
strongest and significant.
**Table 1.** Meteorological inputs for simulations with dynamic and prescribed meteorology.

|  | Dynamic | Prescribed |
|---|---|---|
| Meteorology | Simulated within CAM | GEOS-5 reanalysis data |
| Terrestrial changes | [CTR], [LAI], [HTOP], [NOX], [ALL] | [CTR], [LAI], [HTOP], [NOX], [ALL] |

**Table 2.** Experimental design to quantify surface ozone responses to terrestrial changes including leaf area index (LAI), canopy
height, and soil $NO_x$ emission.

|  | [CTR] | [LAI] | [HTOP] | [NOX] | [ALL] |
|---|---|---|---|---|---|
| LAI | Year 2000 | Year 2050 | Year 2000 | Year 2000 | Year 2050 |
| Canopy height | Year 2000 | Year 2000 | Year 2050 | Year 2000 | Year 2050 |
| Soil $NO_x$ | Year 2000 | Year 2000 | Year 2000 | Year 2050 | Year 2050 |

**3 Year-2000 vs. year-2050 NH₃ emissions and nitrogen deposition**
We first show the year-2000 emissions of reactive nitrogen as $NO_x$ (48 Tg N yr$^{-1}$, Fig. 2a) and $NH_3$ (53 Tg N yr$^{-1}$, Fig. 2b),
with a global budget of 101 Tg N yr$^{-1}$, in good agreement with Ciais et al. (2013). $NO_x$ is densely emitted from industrial and
populated regions, while hotspots for $NH_3$ emission are India and eastern China with intensive agricultural activities and
inefficient fertilizer use. Global year-2050 $NH_3$ emission is projected to reach 67, 57, 65 and 71 Tg N yr$^{-1}$ in Representative
Concentration Pathway (RCP) RCP2.6, RCP4.5, RCP6.0 and RCP8.5 respectively, mainly due to rising agricultural production
(RCP database version 2.0.5). Yet RCP projections did not include a sufficient representation of the spatial patterns of
agricultural $NH_3$ emissions worldwide and especially in Asia, the world's most productive croplands (RCP database version
2.0.5). To capture 2000-to-2050 agricultural intensification, we therefore estimated future $NH_3$ emission based on FAO 2000-
to-2050 crop production changes.

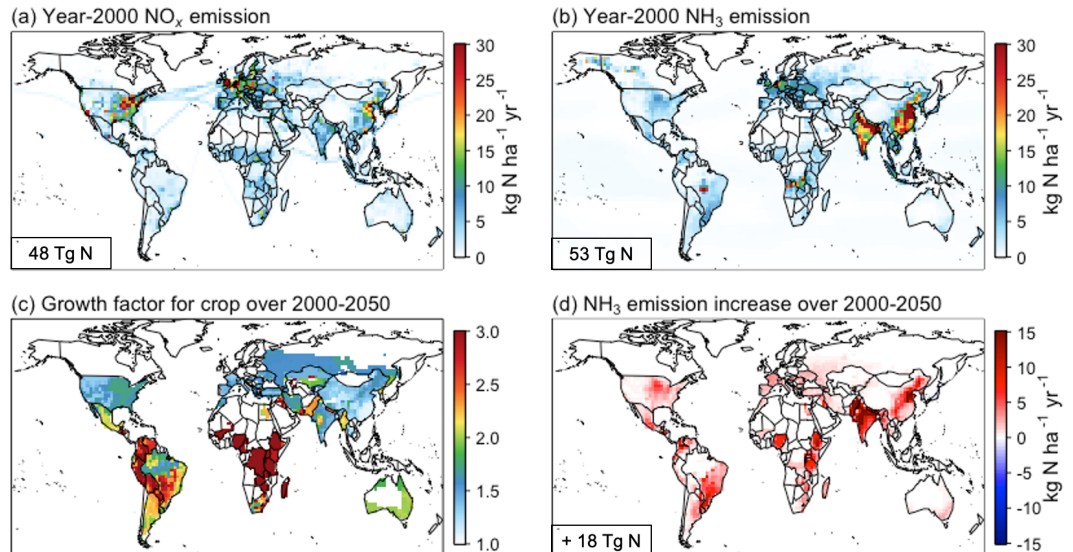

**Figure 2.** Global year-2000 emissions of **(a)** $NO_x$ and **(b)** $NH_3$, **(c)** growth factor $g$ of crop production increase over 2000–
2050 from the Food and Agriculture Organization of the United Nations (FAO), and **(d)** projected increases in $NH_3$ emission
over 2000–2050.
FAO projects global year-2050 crop production to be higher than year-2000 level due to changes in yield, crop intensity (i.e.,
multiple cropping, shortening of fallow periods), and arable land (Alexandratos and Bruinsma, 2012). The major increases
occur in South America and Central Africa due to yield increases and arable land expansion. Production growth factor $g$ in
Fig. 2c can go up to 2–3 for South America and 3–5 for Central Africa, while it is 1.5–2 for some of the world's most productive
croplands at northern midlatitudes, suggesting that the Southern Hemisphere will be playing an increasingly important role in
producing food for the future global population. By scaling up year-2000 $NH_3$ emission by growth factor in FAO crop
production, we estimated that year-2050 $NH_3$ budget to be 71 Tg N yr$^{-1}$, a 34% increase (18 Tg N yr$^{-1}$) compared to year-2000
emission, with major increases over East China, India, Midwestern United States, Brazil, Argentina and East Africa (Fig. 2d).
We fed both year-2000 and year-2050 $NH_3$ emissions into the CESM model to simulate the corresponding nitrogen deposition.
Global budget of both reduced ($NH_x$) and oxidized ($NO_y$) nitrogen deposition is 101 Tg N yr$^{-1}$ in year 2000 (Fig. 3a), which
almost balances out the emission totals of both $NH_3$ and $NO_x$. Nitrogen deposition in 2050 is 121 Tg N yr$^{-1}$, a 20% (20 Tg N
yr$^{-1}$) increase from the year-2000 total (Fig. 3b). Increases in 2000-to-2050 nitrogen deposition mostly result from increased
$NH_x$ deposition, since we fixed the $NO_x$ emission at year-2000 level to isolate the deposition changes due to agricultural
intensification alone. These increased nitrogen deposition serves as an important input of mineral nitrogen from the atmosphere
to the biosphere.

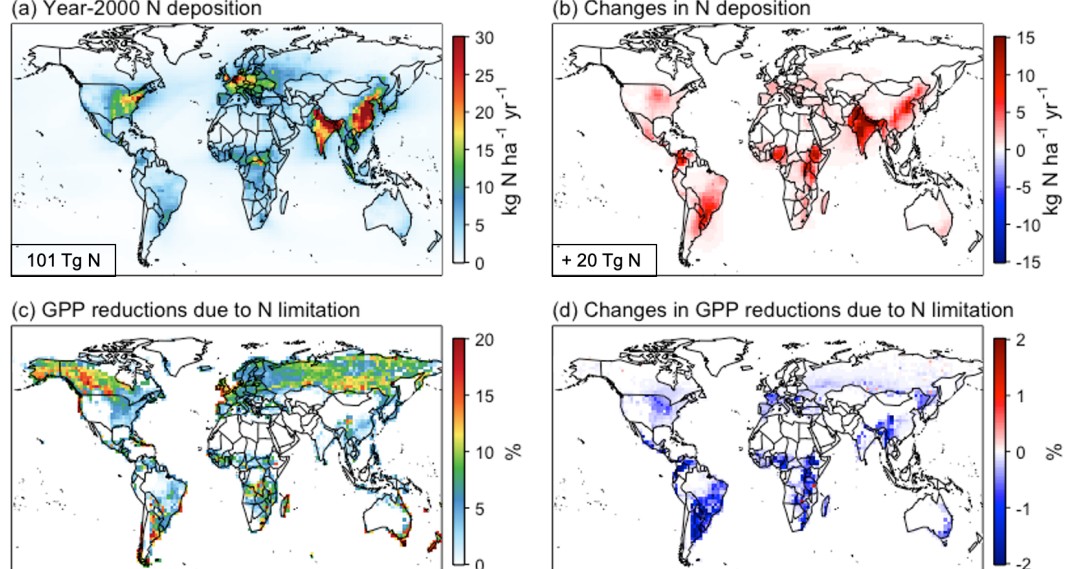

**Figure 3. (a)** Year-2000 atmospheric nitrogen deposition and **(b)** increases in nitrogen deposition over 2000–2050. **(c)** Year-
2000 gross primary production (GPP) reduction due to nitrogen limitation as presented in the CLM model. In nitrogen-limited
soils (i.e., colored areas), plant growth is limited by insufficient soil nitrogen supply due to plant-microbe competition. **(d)**
Changes in nitrogen limitation-induced GPP reductions because of enhanced nitrogen availability from atmospheric nitrogen
deposition over 2000–2050.
**4 Responses of terrestrial ecosystems to nitrogen deposition**
We present in this section the fertilization effect of year-2050 nitrogen deposition and associated enhancements in vegetation
structure (i.e., LAI and canopy height) and soil $NO_x$ emission compared with those of year-2000 nitrogen deposition. Nitrogen
uptake from the soil is an important determinant of plant growth, as nitrogen is a major component of chlorophyll (i.e.,
pigments absorbing light energy for photosynthesis) and Rubisco (i.e., enzyme necessary for carbon fixation). Meanwhile,
mineral nitrogen availability is also vital for nitrification and denitrification microbial processes where $NO_x$ is produced as a
by-product. Yet soil mineral nitrogen, either ammonium ($NH_4^+$) or nitrate ($NO_3^-$), is not always enough to fully meet the
nitrogen demands from both plants and microbes. Here, "nitrogen limitation" is used to describe the nitrogen insufficiency for
plants and microbial uptake. When the soil is "nitrogen-limited", the plants are not able to take up enough nitrogen for
maximum photosynthesis and unmet plant nitrogen demand is translated back to a carbon supply surplus which is eliminated
through reduction of GPP in the CLM model (Fig. 3c). Most of the nitrogen-limited soils are found over the boreal forests
because of slow soil decomposition and turnover with litter of high C:N content and cold climate. Savannas and grasslands in
the tropics are also mildly nitrogen-limited because of low foliar nitrogen concentrations and plant density. Increases in year-
2050 nitrogen deposition (Fig. 3b) lead to more soil nitrogen, enhanced plant growth and thus less GPP reductions in nitrogen-
limited regions (Fig. 3d). However, the nitrogen fertilization effect is not found over nitrogen-abundant regions such as India
and northern China where the critical nitrogen loads are almost always exceeded (Zhao et al., 2017), despite of substantial
increases of nitrogen deposition over 2000–2050.

Due to nitrogen fertilization, GPP, LAI, canopy height and soil $NO_x$ emission over nitrogen-limited regions are generally
higher with year-2050 nitrogen deposition (Fig. 4). Specifically, we found that year-2050 nitrogen deposition to the land



enhances global GPP by 2.1 Pg C yr$^{-1}$ (Fig. 4e), and the enhanced carbon assimilation can be translated into changes in the
carbon mass allocated to different plant parts such as leaves, stems and roots. The two vegetation structural proxies in the
CLM model, LAI and canopy height, which characterize the carbon allocation to plant tissues leaf and stem, respectively. LAI
was simulated to be higher by up to 0.3–0.4 m$^2$ m$^{-2}$ over tropical grasslands and croplands in Brazil, savannas in Sub-Saharan
Africa, and 0.1–0.2 m$^2$ m$^{-2}$ across boreal and temperate forests at midlatitudes (Fig. 4f). Canopy heights from broadleaf
deciduous trees and needleleaf evergreen trees were simulated to be higher by up to 0.1–0.3 m over the eastern US, southern
Europe, southern Russia and southeastern China, and increases of 0.3–0.4 m were found over broadleaf deciduous trees in
South America, and ~0.1 m increases were found for grasses and crops over Sub-Saharan Africa (Fig. 4g). Meanwhile, global
soil NO$_x$ emission budget rises from 7.9 Tg N yr$^{-1}$ to 8.7 Tg N yr$^{-1}$ (Fig. 4h) due to faster and greater nitrification and
denitrification processes under year-2050 atmospheric nitrogen deposition.

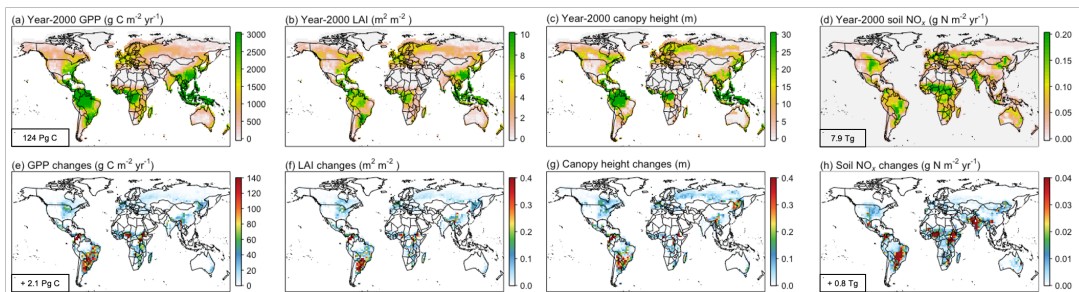

**Figure 4.** Annual mean of year-2000 **(a)** gross primary production (GPP), **(b)** leaf area index (LAI), **(c)** canopy height, **(d)** soil
NO$_x$ emission, and corresponding increases **(e-h)** due to increased nitrogen deposition over 2000–2050.
**5 Impacts of terrestrial changes on surface ozone air quality**
**5.1 Surface ozone changes with prescribed meteorology**
We first examined the responses of surface ozone air quality to changes in LAI, canopy height and soil NO$_x$ separately, as well
as the combined effects of all, with prescribed meteorology (i.e., large-scale meteorological responses to these terrestrial
changes are not accounted for in the ozone changes). With prescribed meteorology, the responses of ozone are seen mostly
where the changes in vegetation cover or soil emission take place. Figure 5d shows that LAI modulates surface ozone
biogeochemically (i.e., without perturbing the overlying meteorology) by ±0.5 ppbv depending on the counteracting effects
from enhanced biogenic VOC emission (Fig. 5e) and surface conductance for ozone deposition (Fig. 5d). We estimated a 3.0
Tg yr$^{-1}$ increase in global biogenic isoprene emission (Fig. 5e), a key source of reduced atmospheric hydrocarbons that are the
chief precursors of tropospheric ozone. Yet, rises in dry deposition velocity (Fig. 5f) reduce ozone concentration. The
sensitivity of isoprene emission to LAI is higher than that of dry deposition, rendering the effects of isoprene emission
dominant in northern midlatitude regions with low LAI to begin with (Wong et al., 2018). As shown in Fig. 5g, increased
canopy height decreases ozone by 0.2 ppbv through stronger aerodynamic conductance and thus stronger turbulent exchange
and dry deposition within the surface layer (without the corresponding changes in the overlying boundary-layer meteorology,
however, due to prescribed meteorology). Ozone dry deposition velocity decreases by 0.002–0.004 cm s$^{-1}$, with increased
canopy height in central Africa and the northern US. Figure 5j shows that surface ozone is elevated biogeochemically by 1–3
ppbv in certain low-NO$_x$ equatorial regions due to increased soil NO$_x$ emission. Overall ozone changes with prescribed
meteorology (Fig. 5m) are mostly local and can be explained predominately (80–90%) by biogeochemical effects from soil
NO$_x$ emission.





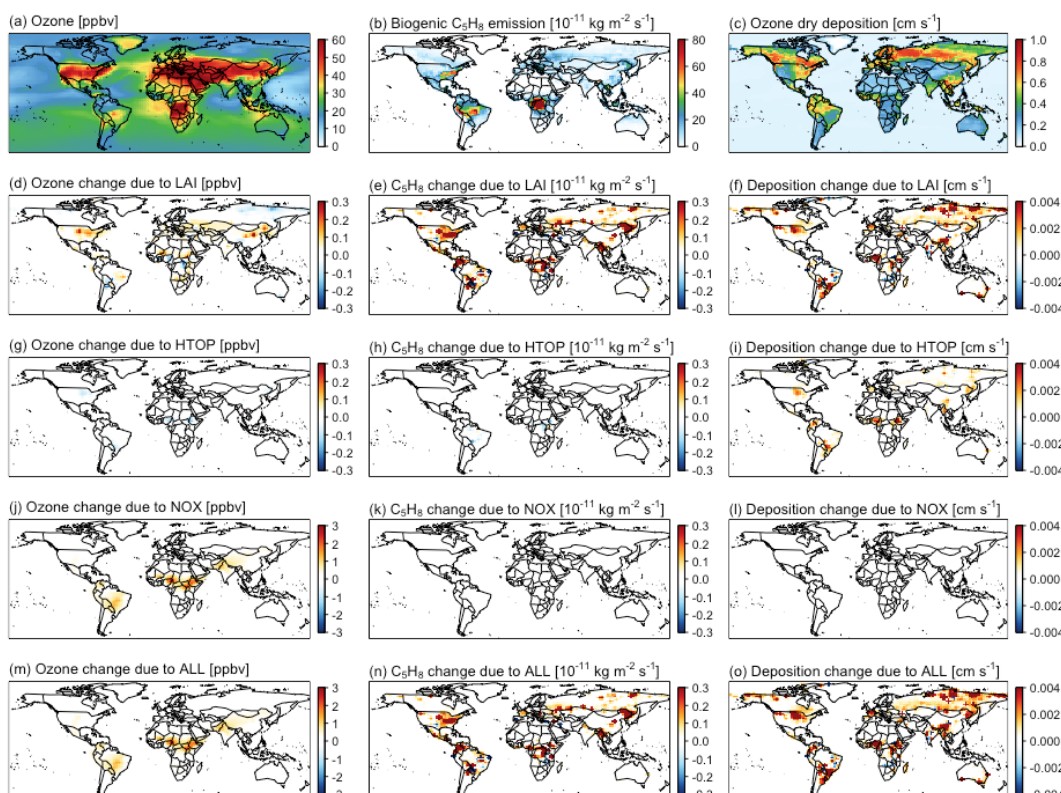

**Figure 5.** Year-2000 summertime (June-July-August; JJA) level of **(a)** surface ozone concentration, **(b)** biogenic isoprene
emission, **(c)** ozone dry deposition, and their corresponding changes due to nitrogen-mediated increases in LAI only **(d, e, f)**,
canopy height only **(g, h, i)** and soil NO$_x$ emission only **(j, k, l)**, and the combination increases of all **(m, n, o)** with prescribed
meteorology.
**5.2 Surface ozone changes with dynamic meteorology**
To evaluate the relative importance of regional terrestrial changes vs. terrestrial changes with meteorological changes in
regulating surface ozone concentration, we also conducted simulations with dynamic meteorology (i.e., overlying boundary-
layer meteorology and large-scale circulation could respond to terrestrial changes). The ozone changes with dynamic
meteorology are the combined results from regional terrestrial changes and associated meteorological changes, an integration
over both biogeochemical and biogeophysical effects. Figure 6 shows that the changes in summertime surface ozone are within
±2–3 ppbv with dynamic meteorology. Overall ozone change with dynamic meteorology (Fig. 6m) are the combined results
from the integrated effects of vegetation changes (Fig. 6d, g) as well as biogeochemical effects of soil NO$_x$ changes (Fig. 6j).
Ozone changes in response to vegetation changes with dynamic meteorology (Fig. 6d, g) are much higher than those with
prescribed meteorology (Fig. 5d, g) as vegetation changes could modify boundary-layer meteorology, shift circulation patterns
and moisture flows, and thus shape ozone concentrations. In contrast to the clear, localized signals in ozone changes through
the biogeochemical pathways, both local and remote surface ozone changes are found when biogeophysical pathways are
involved (Wang et al., 2020). For example, changes in biogenic VOC emission with dynamic meteorology correlate with air





temperature changes (Fig. S1, S2) apart from local vegetation changes. Changes in dry deposition also correlate to
meteorological changes; stomatal resistance can respond to atmospheric dryness and soil water stress (Fig. S1, S2). Ozone
changes in response to soil $NO_x$ changes with dynamic meteorology (Fig. 6j) are within the same magnitude as those with
prescribed meteorology (Fig. 5j), as soil $NO_x$ emissions only change photochemical production of surface ozone, but do not
affect biogenic VOC emission and ozone dry deposition directly (Fig.5 k, l) or via meteorological changes indirectly (Fig.6 k,
l).

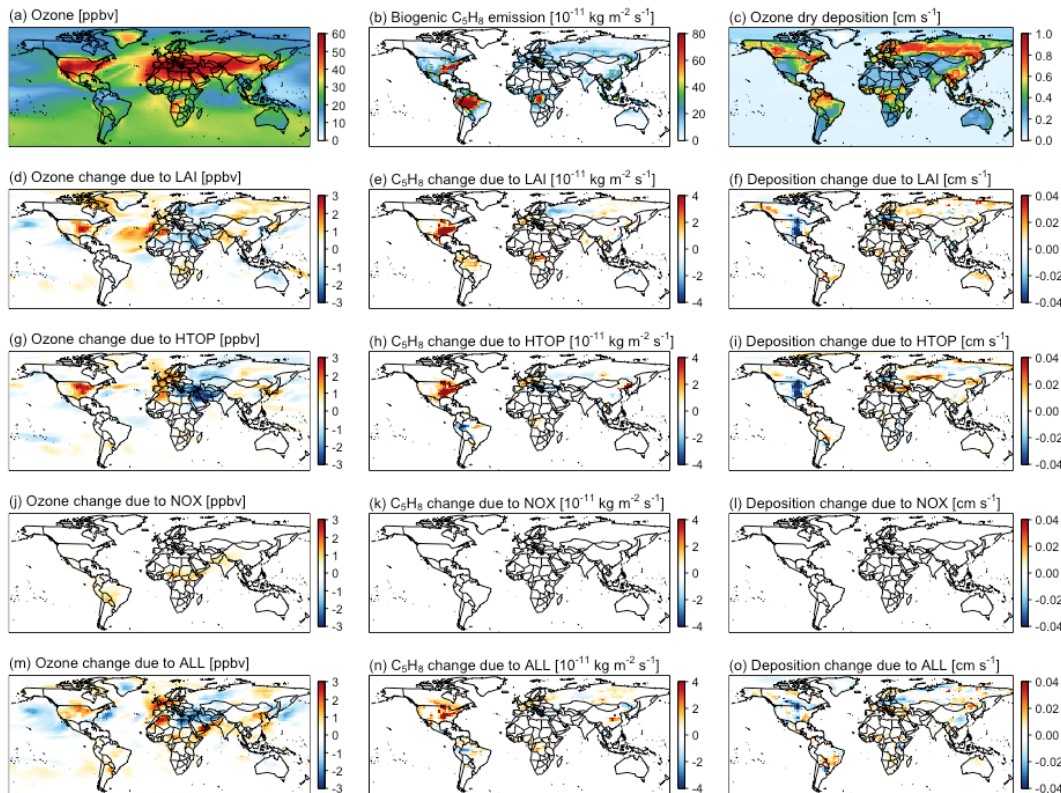

**Figure 6.** Same as Fig. 5 but with dynamic meteorology.
The greatest vegetation enhancements in response to future nitrogen deposition in this study are found over tropical savannas
and grasslands, which are less capable of affecting local and pan-regional climate than forests, and our forest structural changes
are only mild. Therefore, here we choose the US, which shows obvious ozone enhancement following vegetation changes, as
an example to illustrate the biogeophysical effects further. Figure 7 shows that mild LAI increases of temperate deciduous
trees and boreal evergreen trees in the northern US increase local albedo, which results in enhancement in absorbed solar
radiation. The changes in absorbed solar radiation there shift the surface energy balance and circulation patterns in a way that
reduces the moisture convergence in the originally drier places (e.g., the central US, which experiences the greatest ozone
changes), constituting a feedback loop in those less vegetated regions that reduces transpiration, increases temperature,
increases aridity and thus the plant stomata close more (Fig. 7). Yet, our mild vegetation changes have modest local impacts
in places with dense vegetation to begin with (e.g., the eastern US). We found that vegetation changes shift the circulation
patterns and moisture convergence such that it is the adjacent places that are the most affected, which was also found by Wang
et al. (2020), who found obvious temperature increases in the central US after reforestation in the eastern US under RCP4.5



land use and land cover change. High temperature and reduced stomatal conductance in the central US further cause reduced
ozone deposition (Fig. 6f), while increased temperature and LAI in the eastern US enhances biogenic emissions, both of which
increase surface ozone in the central-eastern US (Fig. 6d).

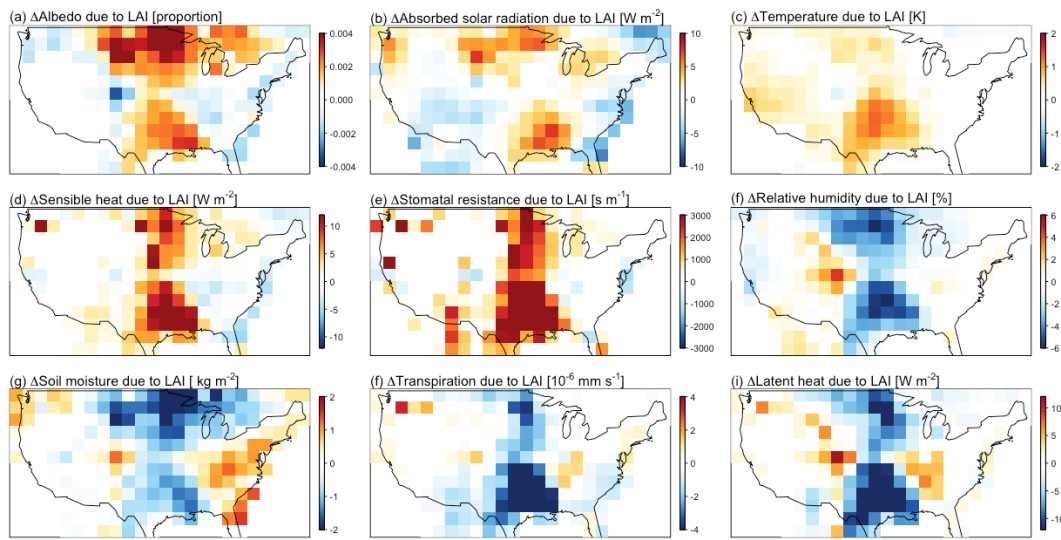

**Figure 7.** Summertime changes in **(a)** absorbed solar radiation, **(b)** albedo, **(c)** 2-meter surface temperature, **(d)** 2-meter relative
humidity, **(e)** soil moisture, **(f)** vegetation transpiration, **(g)** stomatal resistance, **(h)** sensible heat flux, and **(i)** latent heat flux
driven by LAI increase with dynamic meteorology.

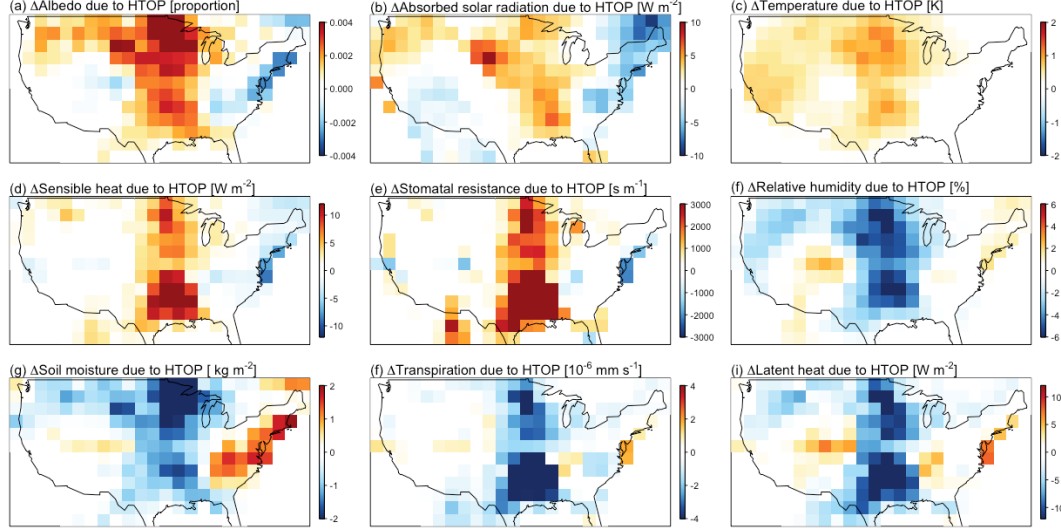

**Figure 8.** Same as Fig.7 but driven by canopy height increase.
Changes in canopy height show similar trends in modulating meteorological conditions (Fig. 8). The effects of meteorological
variations induced by vegetation changes can be as important as or even more important than the direct biogeochemical effects
of vegetation structural changes per se in terms of modulating surface ozone, and are of similar magnitude to the



biogeochemical effects of soil $NO_x$ changes. We note specifically that temperature changes resulted from vegetation-
meteorology coupling are more important than LAI changes per se in regulating biogenic isoprene emission, especially in
regions where obvious warming or cooling occurs.
**6 Conclusions**
With the rising food need for the future world population, more intense agricultural activities are expected to cause substantial
perturbations to the global nitrogen cycle, aggravating surface air pollution and imposing stress on terrestrial ecosystems.
Much less studied, however, is how the ecosystem changes induced by agricultural nitrogen deposition may modify biosphere-
atmosphere exchange and further exert secondary effects on global air quality. In this paper we present a study to quantify the
response of surface ozone air quality to vegetation structural (LAI and canopy height) and soil $NO_x$ emission changes under
year-2000 vs. year-2050 agricultural ammonia emissions over centurial timescales by using an asynchronously coupled
framework.
Agricultural ammonia emission in the coming decades is destined to increase. We estimated year-2050 $NH_3$ emission to be 71
Tg N $yr^{-1}$, a 34% increase compared to year-2000 emission. Our estimate is comparable to 71 Tg N $yr^{-1}$ made by RCP8.5 as
both studies assumed a business-as-usual scenario where future NUE in agroecosystems is not expected to be improved much.
However, it should be acknowledged that increases in food production may also be obtained with a less-than-proportionate
increase in fertilizer use as countries are developing greater awareness of agriculture-related environmental impacts, and
adopting more efficient nutrient use practices in the coming decades. Gu et al. (2015) reported that reasonable changes in diet,
NUE, and N recycling could reduce year-2050 N losses and anthropogenic reactive nitrogen creation to 52% and 64% of 2010
levels, respectively, in China. Fung et al. (2019) showed that the maize-soybean intercropping improves NUE by easing
fertilizer application and $NH_3$ volatilization in agricultural soils in China. Therefore, we acknowledge that the future paths of
agricultural $NH_3$ emission and nitrogen deposition may differ from what we projected as a worst-case scenario in this study,
but we do not expect the nature of the mechanisms and conclusions in this study to be altered significantly.

Atmospheric nitrogen deposition increases carbon uptake by terrestrial biosphere in nitrogen-limited areas, and also stimulates
release of $NO_x$, nitrous oxide ($N_2O$) and $NH_3$ from soils (Reis et al., 2009; Zaehle et al., 2011). We found that nitrogen
deposition increases by 20% from year 2000 to 2050 due to rising agricultural $NH_3$ emission, and this enhances global GPP
by 2.1 Pg C $yr^{-1}$. LAI was simulated to be higher by up to 0.3–0.4 $m^2\ m^{-2}$ in tropical grasslands and croplands, and 0.1–0.2
$m^2\ m^{-2}$ in midlatitude boreal and temperate forests. Canopy height increases were found in boreal and temperate forests (by
0.1–0.4 m), as well as in tropical grasslands and croplands (by ~0.1 m). Soil $NO_x$ emission budget rises to 8.7 Tg N $yr^{-1}$ with
year-2050 nitrogen deposition because of intensive nitrification and denitrification processes. Due to decreasing trends of
anthropogenic $NO_x$ emission throughout this century (IPCC, 2013), soil $NO_x$ is expected to play an increasingly important role
in global $NO_x$ budget. Therefore, the inclusion of effects of soil $NO_x$ emission to surface ozone is essential. These estimates
are based on carbon and nitrogen interactions in CLM4.5 biogeochemistry (CLM4.5-BGC), which are widely used in
estimating long-term trajectory of terrestrial variations (Lombardozzi et al., 2012; Val Martin et al., 2014; Sadiq et al., 2017;
Zhou et al., 2018). However, the internal soil nitrogen cycle, its coupling with the atmosphere and reactive nitrogen gas
emissions other than $N_2O$ are not fully represented in default CLM4.5-BGC. The soil $NO_x$ emission module that we added,
which allows soil $NO_x$ to respond to nitrogen deposition from the atmosphere, partly improved the representation (Fung et al.,
2021), but the $NH_3$ emission we used was still based on inventories and scaling with future crop production and thus did not
respond to nitrogen deposition. We expect, however, that the secondary effect of nitrogen deposition on $NH_3$ should be much





smaller than any perturbations due to agricultural changes (Fung et al., 2021). Moreover, fully coupled bidirectional nitrogen
fluxes were not enabled in our model setting. Future work is needed to examine the overall downstream biogeochemical and
biogeophysical effects in an Earth system model with a closed nitrogen cycle where soil $NO_x$ and $NH_3$ emissions to the
atmosphere and nitrogen deposition from the atmosphere are fully coupled dynamically.
With only the biogeochemical effects of nitrogen-induced terrestrial changes (with prescribed meteorology where
meteorological changes are not included), surface ozone is elevated by 1–3 ppbv in certain low-$NO_x$ equatorial regions due to
increased soil $NO_x$ emission, while LAI and canopy height only modulate surface ozone by ±0.5 and 0.2 ppbv, respectively.
With both the biogeochemical and biogeophysical effects under dynamic meteorology, changes in summertime surface ozone
are within ±2–3 ppbv. Ozone responses due to vegetation changes are much higher with dynamic meteorology than prescribed
meteorology, as vegetation changes shift surface energy balance, circulation patterns, moisture flow, and thus shape ozone
concentrations. Local meteorological variations induced by vegetation structural changes are generally more important than
the vegetation changes per se in terms of modulating surface ozone concentration, and appear to be as important as
biogeochemical soil $NO_x$ effect. Furthermore, biogeophysical pathways related to canopy height changes have not been
accounted for by most previous studies of ozone-vegetation interactions, which usually only considered LAI and other
ecophysiological changes (Wang et al, 2020; Wong et al., 2018; Zhao et al., 2017; Fu et al., 2015). Global vegetation growth
is altered by land use and land cover change, warming, $CO_2$ fertilization, nitrogen deposition and ozone damage, etc., but the
associated canopy height changes have usually been ignored, rendering an incomplete representation of terrestrial effects on
surface air quality predictions. Here, we found that the effects of canopy height changes on surface ozone through the
biogeophysical pathways are noticeable and can be as much as the effects associated with LAI changes alone.
Overall, our study demonstrates a novel linkage between agricultural activities and ozone air quality via the modulation of
vegetation and soil biogeochemistry by nitrogen deposition, and highlights the particular importance of considering
meteorological changes following vegetation structural changes including those in canopy height, as well as soil $NO_x$ changes,
in studying the effects of ozone-nitrogen-vegetation interactions in the future.

**Acknowledgement**
This work was supported by Research Grants Council (RGC) General Research Fund (Reference #: 14323116) and National
Natural Science Foundation of China (NSFC)/RGC Joint Research Scheme (Reference #: N_CUHK440/20) awarded to A. P.
K. Tai.

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
