# Peer review of "and subsequent nitrogen deposition through terrestrial ecosystem"

_Atmospheric Chemistry and Physics, 2021_

## Referee Comment (RC2)

Comments on:

**Responses of surface ozone to future agricultural ammonia emissions and subsequent nitrogen deposition through terrestrial ecosystem changes**

Liu et al.,

submitted to Atmospheric Chemistry and Physics, August 2021

Decision: accept with minor revision and clarification

**General comments**:

In this manuscript, authors present a novel linkage between agricultural activities and ozone air quality, by examining the responses of surface ozone air quality to terrestrial changes caused by 2000-to-2050 increased ammonia emission and resulted increased nitrogen deposition. Authors make use of CESM model to investigate each individual and combined effects of LAI, canopy height and soil NOx, and try to isolate biogeochemical effects by using prescribed meteorology. In general, the manuscript is very well written! I think this manuscript meets the criteria for publication on Atmospheric Chemistry and Physics:

- It is an advancement in understanding the linkage between ozone air quality and agricultural activities.

- Evidence provided by the authors are strong for the conclusion drawn

- This work is of importance to researchers studying atmospheric chemistry, physics and atmosphere-biosphere interactions

However, there are some questions and details needed to be further addressed from my perspective:

**General revision suggestions**:

- Figure 1. This illustration is very helpful to readers who are not very familiar with the complex interactions between atmospheric chemistry and terrestrial ecosystem. Since one of the major conclusions is that ozone changes are typically larger when meteorology is dynamically simulated, I am wondering whether some biogeophysical effects/pathways could be added to this diagram. I understand it could get

overcomplicated very fast, but maybe one or two pathways explained in Figure 7 should be added.

- Spin-up period for the model. I see that CLM45BGC mode has been spun-up for 150 years, and then 50 years for steady state. Perturbation experiment is then done for another 60-70 years. This seems an impressively long period of time for spin-up and perturbation. Is this a common practice for this mode of CLM model? Or how did you determine that the model has reached a steady state? Did the model start from zero vegetation (LAI=0)? I am interested to look at maybe just one figure showing the evolution of mean LAI over certain region during these hundreds of years of simulation. You don't have to include it in the appendix.

- P7L12, '..., we estimated that year-2050 NH3 budget to be 71 Tg N yr$^{-1}$, ...'. I noticed and you discussed later as well that this number is the same as RCP8.5 projection. It is probably worth mentioning the fact and that FAO makes similar assumption as RCP 8.5 scenario here.

- Figure 3b. I think it would be more beneficial to have this figure in percentage changes rather than absolute changes.

- In Figure 3c, you have shown GPP reduction due to nitrogen limitation. I noticed some discussion about it is given in Section 4. However, I am wondering how you obtained this variable. I might have missed the part where you introduce this, but did you compute it by comparing two simulations (one with and the other without nitrogen limitation), or is it from some nitrogen limitation parameter in the model? Some introductions could be added in Section 3 or 4.

Also, some **technical corrections** need to be made before the publication:
- P3L4, 'facilities' to 'facilitates'.
- Figure 7 and 8, labels are inconsistent between caption and subpanels. Also, there are two subpanels labelled f.

---

## Author Comment (AC1)

**Response to comments of Referee #1**

**Manuscript number:** acp-2021-492
**Authors:** Xueying Liu, Amos P. K. Tai, Ka Ming Fung
**Title:** Responses of surface ozone to future agricultural ammonia emissions and subsequent nitrogen deposition through terrestrial ecosystem changes

*This study provides a very interesting modelling study of the potential global impacts of changing agricultural demand, and thus ammonia emissions, on future surface ozone concentrations. The study provides a comprehensive set of scenarios looking at different vegetation responses to increased fertiliser use on ozone concentrations between 2000 and 2050, using both prescribed and dynamic meteorology. Overall it was shown that increased nitrogen fertiliser use by 2050 leads to increased LAI and thus enhanced surface ozone concentrations, with the biggest impact seen when dynamic meteorological affects were allowed. This study is suitable for publication in ACP after the following comments are addressed.*

> We would like to thank you for the thoughtful and insightful comments. The manuscript has been revised accordingly, and our point-by-point responses are provided below. The reviewer' comments are *italicized*, our new/modified text cited below is highlighted in **bold**. The revised manuscript with tracked changes is also included at the end for easy reference.

*Specific Comments*
*Does this version of the CLM include the impacts and feedbacks of ozone damage on stomatal resistances? If so were they included in the simulations preformed in this work? This could potentially have further impacts on the nitrogen deposition effects on vegetation, particularly through stomatal uptake.*

> The reviewer aptly pointed out that ozone damage on stomatal conductance could affect nitrogen deposition through stomatal uptake. We have now addressed it in P15 L7 that **"One limitation of this study is that we did not consider ozone damage on stomatal conductance and photosynthesis as in the study by Sadiq et al. (2017). If ozone damage on stomatal conductance is considered, higher ozone concentrations could have positive feedbacks on ozone itself via reduced dry deposition and enhanced isoprene emission. Meanwhile, ozone damage on plant productivity may also diminish the fertilization effect of nitrogen and foliar nitrogen content, which is itself vital for photosynthetic capacity (Franz and Zaehle, 2021). Therefore, if ozone damage is considered, lower LAI and canopy height are expected, compensating some of the enhanced LAI and canopy height induced by higher nitrogen deposition found in this study. These changes in LAI and canopy height could further affect ozone via various biogeochemical and biogeophysical pathways, but such a secondary feedback effect is expected to be relatively minor (Zhou et al., 2018). More work is warranted to investigate the individual and combined effects of nitrogen deposition and ozone damage on plant growth and terrestrial carbon uptake, especially in light of the**

**possible nonlinear interactions between ozone and nitrogen in plants (e.g., Shang et al., 2021).**"

*In Section 5 the authors present a very good summary of the potential feedbacks caused by changes in nitrogen deposition in response to future changes in agricultural practices. In particular they focus on the feedbacks through changes in LAI and canopy height. However, they do not cover the potential feedbacks involved where changes in ozone concentrations could lead to plant damage and thus impacts on not only ozone concentrations themselves but also uptake of nitrogen species. It is appreciated that given the current setup of the modelling system a further simulation is not possible but would the authors be able to give a more detailed comparison with the potential effects of ozone damage on the results observed or comment on how this could affect the results simulated by the model.*

Please see above response.

*Technical Comments*
*Page 1, Line 24: Please change to emissions*

Revised as suggested.

*Page 2, Line 4: The start of this sentence seems a little repetitive, please correct to something like 'Crops typically take up only about 40-60% of the nitrogen fertiliser applied…….'*

Revised as suggested.

*Page 5, Line 13: Do you mean Fig 2 here?*

Revised as suggested.

**Response to comments of Referee #2**

**Manuscript number:** acp-2021-492
**Authors:** Xueying Liu, Amos P. K. Tai, Ka Ming Fung
**Title:** Responses of surface ozone to future agricultural ammonia emissions and subsequent nitrogen deposition through terrestrial ecosystem changes

*Comments on:*
*Responses of surface ozone to future agricultural ammonia emissions and subsequent nitrogen deposition through terrestrial ecosystem changes*
*Liu et al., submitted to Atmospheric Chemistry and Physics, August 2021*
*Decision: accept with minor revision and clarification*

*General comments:*
*In this manuscript, authors present a novel linkage between agricultural activities and ozone air quality, by examining the responses of surface ozone air quality to terrestrial changes caused by 2000-to-2050 increased ammonia emission and resulted increased nitrogen deposition. Authors make use of CESM model to investigate each individual and combined effects of LAI, canopy height and soil NOx, and try to isolate biogeochemical effects by using prescribed meteorology. In general, the manuscript is very well written! I think this manuscript meets the criteria for publication on Atmospheric Chemistry and Physics:*
*- It is an advancement in understanding the linkage between ozone air quality and agricultural activities.*
*- Evidence provided by the authors are strong for the conclusion drawn*
*- This work is of importance to researchers studying atmospheric chemistry, physics and atmosphere-biosphere interactions*
*However, there are some questions and details needed to be further addressed from my perspective:*

> We would like to thank you for the thoughtful and insightful comments. The manuscript has been revised accordingly, and our point-by-point responses are provided below. The reviewer' comments are *italicized*, our new/modified text cited below is highlighted in **bold**. The revised manuscript with tracked changes is also included at the end for easy reference.

*General revision suggestions:*
*Figure 1. This illustration is very helpful to readers who are not very familiar with the complex interactions between atmospheric chemistry and terrestrial ecosystem. Since one of the major conclusions is that ozone changes are typically larger when meteorology is dynamically simulated, I am wondering whether some biogeophysical effects/pathways could be added to this diagram. I understand it could get overcomplicated very fast, but maybe one or two pathways explained in Figure 7 should be added.*

> We have now added biogeophysical pathways in Figure 1 and its caption: "Figure 1. "Biogeochemical" **and "biogeophysical"** pathways of nitrogen deposition affecting surface ozone concentration. Biogeochemical pathways via canopy height (yellowcolored), leaf area index (LAI; green-colored), and soil $NO_x$ (blue-colored), **as well as some of the biogeophysical pathways relevant for this study (red-colored)** are shown. The sign associated with each arrow indicates the correlation between the two variables; the sign of the overall effect (positive or negative) of a given pathway is the product of all the signs along the pathway. "Biogeochemical" pathways affect gas exchange (i.e. biogenic VOC emission and ozone deposition) though plant stomata or microbe-mediated soil processes. **"Biogeophysical" or "meteorological" pathways are mediated through a modification of the local and nonlocal overlying meteorological environment above the surface layer**."

*Spin-up period for the model. I see that CLM45BGC mode has been spun-up for 150 years, and then 50 years for steady state. Perturbation experiment is then done for another 60-70 years. This seems an impressively long period of time for spin-up and perturbation. Is this a common practice for this mode of CLM model? Or how did you determine that the model has reached a steady state? Did the model start from zero vegetation (LAI=0)? I am interested to look at maybe just one figure showing the evolution of mean LAI over certain region during these hundreds of years of simulation. You don't have to include it in the appendix.*

The 200-year simulation was to provide a steady-state initial condition for the perturbation experiments later. It started from the default initial condition files with certain LAI values (see right panel of Figure R1). We wanted to make sure that the LAI was stabilized at year-2000 level, so looping over year-2000 for 200 simulation years was adopted. The same practice is also used in Sadiq et al. (2017), Zhou et al. (2018), and Wang et al. (2020). After this, the actual perturbation experiments were simulated for 70 years. Figure R2 shows the LAI differences between year-2000 and year-2050 for the first 10–20 years. For all four regions, we observed the LAI differences are stabilized within the first 10–20 years, and then averaged the remaining 50 years as year-2050 steady state.

We have now explained further in P5 L34 that "**We used the year-2000 steady state as initial conditions for the following perturbation experiments.** We then perturbed the present-day steady state with future nitrogen deposition fluxes following the year-2050 agricultural emission scenario, allowing the vegetation and soil variables to come into a "new" steady state, which took 10–20 simulations years. After that, the simulation was conducted for another 50 years, which were **considered to be year-2050 steady state and then** averaged to determine the differences in LAI, canopy height and soil $NO_x$ emission from the 50-year present-day averages."

[Figure]

Figure R1. Left panel shows mean LAI of the 200-year simulation, and right panel shows LAI evolution of South America (red box in left panel).

[Figure]

Figure R2. The LAI differences between year-2000 and year-2050 for the first 10–20 simulation years.

Reference:

Sadiq, M., Tai, A. P. K., Lombardozzi, D., and Val Martin, M.: Effects of ozone-vegetation coupling on surface ozone air quality via biogeochemical and meteorological feedbacks, Atmos. Chem. Phys., 17, 3055–3066, https://doi.org/10.5194/acp-17- 3055-2017, 2017.

Zhou, S. S., Tai, A. P. K., Sun, S., Sadiq, M., Heald, C. L., and Geddes, J. A.: Coupling between surface ozone and leaf area index in a chemical transport model: strength of feedback and implications for ozone air quality and vegetation health, Atmos. Chem. Phys., 18, 14133–14148, https://doi.org/10.5194/acp-18-14133-2018, 2018.

Wang, L., Tai, A.P., Tam, C.Y., Sadiq, M., Wang, P. and Cheung, K.K.,: Impacts of future land use and land cover change on mid-21st-century surface ozone air quality: distinguishing between the biogeophysical and biogeochemical effects, Atmos. Chem. Phys., 20, 11349–11369, https://doi.org/10.5194/acp-20-11349-2020, 2020.

*P7L12, '..., we estimated that year-2050 NH3 budget to be 71 Tg N yr-1, ...'. I noticed and you discussed later as well that this number is the same as RCP8.5 projection. It is probably worth mentioning the fact and that FAO makes similar assumption as RCP 8.5 scenario here.*

We have now mentioned this in P7 L14 that "**This estimate is comparable to the RCP8.5 estimate of 71 Tg N yr$^{-1}$ as both studies assumed a business-as-usual scenario where future NUE in agroecosystems is not expected to be improved much.**"

*Figure 3b. I think it would be more beneficial to have this figure in percentage changes rather than absolute changes.*

We are happy to show the percentage changes of nitrogen deposition over 2000–2050. The current setting of Figure 3 is year-2000 nitrogen deposition and percentage GPP reduction on the left, and the absolute differences by year-2050 minus year-2000 on the right. If we change absolute difference in panel (b) to percentage difference, we would also need to change panel (d) to percentage difference to be consistent. Yet in this case, panel (d) becomes percentage difference of panel (c) percentage GPP reduction, which is less straightforward and complicates the explanation we show in Sect. 4.

As an alternative, we have now put the percentage changes in supplementary Figure S3, and state in P8 L6 that "**Relative changes over 2000–2050 can be found in supplementary Figure S2.**".

*In Figure 3c, you have shown GPP reduction due to nitrogen limitation. I noticed some discussion about it is given in Section 4. However, I am wondering how you obtained this variable. I might have missed the part where you introduce this, but did you compute it by comparing two simulations (one with and the other without nitrogen limitation), or is it from some nitrogen limitation parameter in the model? Some introductions could be added in Section 3 or 4.*

Nitrogen limitation is from a model output variable called "downregulation", which stands for downregulation of potential carbon allocation based on soil nitrogen availability.

We have now further explained this in P8 L14 that "…**In CLM, the plant nitrogen demand for new growth is calculated by the carbon available for allocation to new growth allocation, given the C:N stoichiometry of a given plant type and plant part. From the soil side, soil mineral nitrogen supply is calculated by adding various nitrogen sources (e.g., atmospheric nitrogen deposition, fertilizer, biological nitrogen fixation) and subtracting nitrogen sinks (e.g., leaching, assimilation by heterotrophs). When the plant nitrogen demand is greater than the soil nitrogen supply, the plants are not able to take up enough nitrogen to support the carbon allocation for new growth, which would then be reduced ("downregulated") by a percentage in the model, which we refer as soil "nitrogen limitation" on plant growth here.** When the soil is "nitrogen-limited", the plants are not able to take up enough nitrogen for maximum photosynthesis and unmet plant nitrogen demand is translated back to a carbon supply surplus which is eliminated through reduction of GPP in the CLM model. **Figure 3c shows**

**the year-2000 GPP percentage reductions due to nitrogen limitation.** Most of the nitrogen-limited soils are found over the boreal forests because of slow soil decomposition and turnover with litter of high C:N content and cold climate. Savannas and grasslands in the tropics are also mildly nitrogen-limited because of low foliar nitrogen concentrations and plant density. **Figure 3d shows the differences of GPP reductions, i.e., year-2050 GPP reductions minus year-2000 GPP reductions. We found smaller GPP reductions induced by nitrogen limitation in 2050 than 2000, reflecting higher plant productivity and growth over 2000–2050. However, this nitrogen fertilization effect is found only over nitrogen-limited regions, but not over** nitrogen-abundant regions such as India and northern China where the critical nitrogen loads are almost always exceeded (Zhao et al., 2017) despite of substantial increases of nitrogen deposition over 2000–2050.

*Also, some technical corrections need to be made before the publication:*
*P3L4, 'facilities' to 'facilitates'.*

Revised as suggested.

*Figure 7 and 8, labels are inconsistent between caption and subpanels. Also, there are two subpanels labelled f.*

Revised as suggested.

**Response to comments of Referee #3**

**Manuscript number:** acp-2021-492
**Authors:** Xueying Liu, Amos P. K. Tai, Ka Ming Fung
**Title:** Responses of surface ozone to future agricultural ammonia emissions and subsequent nitrogen deposition through terrestrial ecosystem changes

*This manuscript presented a modelling study that aimed to quantify how future changes in atmospheric nitrogen deposition as driven by rising agricultural food production affect surface ozone levels via air-biosphere interactions. Asynchronously coupled air-biosphere modelling simulations were conducted using the atmosphere and land components of the Community Earth System Model (CESM), so that the individual biogeoschemical and biogeosphysical pathways of the nitrogen deposition-surface ozone air quality linkage. The results emphasize the importance of biogeophysical pathways or the meteorological variations induced by vegetation changes in modulating surface ozone.*

*The manuscript is overall well conducted and presented. The simulations are well designed, and the analyses identify a new linkage of agricultural nitrogen and air pollution. I suggest publish on ACP after the following comments been addressed.*

We would like to thank you for the thoughtful and insightful comments. The manuscript has been revised accordingly, and our point-by-point responses are provided below. The reviewer' comments are *italicized*, our new/modified text cited below is highlighted in **bold**. The revised manuscript with tracked changes is also included at the end for easy reference.

*Specific comments:*
*1) Page 5, Eq. 1: A few more sentences describing the growth factor are suggested. How it treats different crops? Could it consider ammonia emission factors may be different for different crops? Please clarify.*

We generated growth factors for major crops and obtained an average growth factor from these crop-specific production growths. We have now clarified this in P5 L16 that "…**We generated the growth factors for major crops (Fig. S1) and obtained an average growth factor from these crop-specific production growths.**"

Agricultural NH$_3$ emission rates are different for different crops in the MASAGE_NH3 inventory (Paulot et al., 2014), which stands for year-2000 conditions. We assumed emission rates of each specific crop to remain the same in the future, which can be regarded as a representation of the "worst-case" scenario where fertilizer nitrogen use remains as inefficient as it is today.

*2) Page 5, Line 20: Each atmospheric chemistry simulation was conducted for 20 years. What meteorology fields were used to represent the 2000 and 2050 conditions? Please clarify.*

We have now clarified it further in P5 L21 that "…For each scenario of the sensitivity experiments, CAM-Chem simulations were conducted for 20 simulation years.

**Throughout the CAM-Chem component was coupled online with CLM45SP with prescribed vegetation structures, which computed land-atmosphere fluxes for CAM-Chem to simulate atmospheric dynamics and chemistry. Both simulations were performed with prescribed sea surface temperature and sea-ice cover following the HadISST dataset (Rayner et al., 2003) at the year-2000 level. Long-lived greenhouse gases and their radiative forcing were kept at year-2000 level to exclude the effects of increasing temperature on NH$_3$ emissions.** The first five years…"

There are also more details on meteorological fields in P4 L16: "…CAM-Chem provides the flexibility of performing climate simulations online (i.e., "dynamic meteorology") and simulations with specified meteorological fields (i.e., "prescribed meteorology"). For simulations with dynamic meteorology, it was driven by the Climatic Research Unit – National Centers for Environmental Prediction (CRU-NCEP) climate forcing dataset. For simulations with prescribed meteorology, year-2000 and 2001 horizontal wind components, air temperature, surface temperature, surface pressure, sensible and latent heat flux and wind stress of the Goddard Earth Observing System Model version 5 (GEOS-5) forcing data at six hour interval were used (see Table 1). This version of CAM-Chem…"

*3) Page 9, Line 29: "Ozone dry deposition velocity decreases by 0.002-0.004 …". Should it be increases in ozone dry deposition velocity as shown by figure 5?*

    Revised as suggested.

*4) Page 11, Figure 6: It appears that the individual effects do not add up when with dynamic meteorology. As shown in this figure, ozone changes due to LAI (figure 6d) and due to HTOP (figure 6g) show large positive values in the central US, while the combined effects (due to ALL, figure 6m) become much weaker. The same issue can be seen for deposition velocity changes over the US (figure 6f/i/6o). Can you explain why?*

    We have now attempted to address the issue more in P13 L9: "**It is noteworthy that unlike with prescribed meteorology, individual effects may not add up linearly with dynamic meteorology for a given location due to the complex and far-reaching changes in atmospheric circulation and the associated cascade of local and nonlocal changes in climate that are dynamically simulated following terrestrial changes.**"

*5) Page 11, Line 15: "increase local albedo, which results in enhancement in absorbed solar radiation". It is not clear why higher albedo could lead to higher absorbed solar radiation, as higher albedo tends to reflect more solar radiation back to the atmosphere. Please clarify.*

    We have now revised Figure 7 and also revamped the explanation of the biogeophysical mechanisms behind in P11 L21, which does not involve the questionable changes in albedo anymore: "Therefore, here we choose the US which shows obvious ozone enhancement following vegetation changes, as an example to illustrate the biogeophysical effects further. **Figure 7 shows that in the forest regions in the eastern US where LAI and canopy height changes are relatively large following higher nitrogen deposition, albedo decreases, absorbed radiation increases, latent heat flux increases, and such changes**

**appear to have shifted the surface energy balance and circulation patterns in a way that enhances moisture convergence, precipitation and soil moisture in the originally wetter places (i.e., the forested eastern US), but reduces the moisture convergence in the originally drier places (i.e., the grassland regions in the central US). This constitutes a feedback loop in these grassland regions that reduces transpiration, increases temperature, increases aridity and thus the plant stomata close more, all leading to the relatively large enhancements in surface ozone there.** Our mild vegetation changes **only** have modest local impacts in places with dense vegetation to begin with (e.g., the eastern US). …"

*6) Page 11, Line 22: Wang et al. (2020) is not listed in the References;*

Revised as suggested.

*Page 12, Figure 7: There are two (f) panels in the figure;*

Revised as suggested.

*Page 16: Line 26-31, missing journal and page information for the two citations.*

Revised as suggested.